# Knowledge exchange in the implementation of National Environmental Programmes (NEPs) in China: A complex picture

**Zheng-Hong Kong**[1]*, **Lindsay C. Stringer**[1,2], **Jouni Paavola**[3]

**1** Department of Environment and Geography, University of York, York, North Yorkshire, United Kingdom,
**2** York Environmental Sustainability Institute, University of York, York, North Yorkshire, United Kingdom,
**3** School of Earth and Environment, University of Leeds, Leeds, West Yorkshire, United Kingdom

* zk674@york.ac.uk

**Data Availability Statement:** The data cannot be shared publicly due to ethical considerations. The data are available from the Legal Services, Corporate and Information Services, University of

## Abstract

Knowledge is an intrinsic element of environmental management. Understanding what kinds of knowledge are needed and how to communicate them effectively is crucial for building environmental management capacity. Despite extensive research, knowledge and its exchange are commonly considered from the viewpoint of its creators and disseminators, rather than that of its recipients. This can lead to mismatches between supply of and demand for knowledge, and futile knowledge exchange that undermines the effectiveness of interventions. Research is needed that looks carefully at the contexts and consequences of such scenarios. Addressing this gap, we examine the implementation of National Environmental Programs (NEPs) in north-western China, drawing from interviews and questionnaires with scientists, grassroots implementers, and farmers and herders, to identify what and how knowledge has been exchanged and what their perspectives are about knowledge exchange with other actors. We ascertain the positive impacts of knowledge exchange during NEP implementation, as well as the consequences when it is lacking, by analysing the interfaces and interactions between actors, seeking explanation for successes and failures. We conclude that with changing socio-ecological systems, knowledge and its exchange also need to change accordingly, extending beyond the environmental domain to integrate local socioeconomic concerns. Such efforts are necessary to improve environmental management outcomes and advance sustainable development.

## Introduction

Knowledge exchange (KE) is usually undertaken in environmental management to inform policymakers and invoke social learning, knowledge co-production, and co-management among stakeholders [1–6]. Such KE, i.e., the "processes that generate, share and/or use knowledge through various methods appropriate to the context, purpose, and participants involved" [1], is increasingly recognised as key to facilitating social, environmental, and economic impacts of research, policy and practice. To improve KE between scientists and policymakers, research has to be explicitly and demonstrably policy relevant so that it can provide pathway(s)

York for scientists who meet the criteria for access to confidential data. Data access requests can be sent to the Legal Services, Corporate and Information Services, University of York, Heslington, York, Yorkshire, YO10 5DD. Phone: 01904 323869, Email: dataprotection@york.ac.uk.

**Funding:** The authors received no specific funding for this work.

**Competing interests:** The authors have declared no competing interests exist.

for policy impact. Enabling factors, such as identifying policymakers and their information and knowledge needs, are helpful for making scientific research available, visible, accessible, and compatible with these needs [7]. To encourage the use of research, scientists are advised to incorporate potential users' needs into project plans and ensure their engagement in research activities [7–9]. Effects of different strategies in participatory management have been explored and elucidated [10–16]. While research has rigorously examined the processes for increasing the use of knowledge, the outcomes of knowledge use has been given less attention [17]. From useful knowledge that scientists believe to usable knowledge that users really use, there are many factors and needs interactions at varying levels [18, 19]. Dilling and Lemos suggest (2011) that usability of science is determined by its production process as well as context of its potential use, and successful use of knowledge involves iteration between knowledge producers and users [20]. It is also difficult to ascertain the usability of the knowledge due to the complexities and emergencies arising from the intersections of knowledge production and its use [21].

Current research puts considerable emphasis on scientists as producers of usable knowledge, alongside policy makers and practitioners as the main users, while acknowledging that present environmental challenges pose threats to both social and ecological systems, with actions needed from everyone. Rist et al. (2016) have pointed out that the needs of local communities, such as smallholder farmers and herders, have not been adequately attended to in KE [3]. The needs of street-level bureaucrats, i.e., as Sevä & Jagers put, "the practicing and, typically, anonymous civil servants at the very end of the environmental policy chain" in the top-down system, are also rarely considered in policy arenas [22]. However, these groups work for and are often directly affected by both the policies and the environmental issues that the policies aim to address. Their actions and behaviours determine the effectiveness of policies and therefore they are important but often neglected stakeholders within KE processes.

Environmental management is an engagement of multiple actors with different knowledge backgrounds [3]. Because of the existence of multiple social realities, these different knowledges need to learn from each other [23, 24]. Similarly, complex and dynamic social-ecological systems and processes within which environmental management happens, also require the integration of a diversity of knowledge and values for comprehensive understanding of the systems of interest [25]. At the same time, environmental decisions often require trade-offs to be made when scientific evidence, economic effects, and political priorities are considered together. Through KE, appreciation of varying perspectives to, interests in, and fundamental philosophies regarding the problems of environmental management can be supported, so that conflicts can be dealt with, and incentives for compliance may be devised [26]. Conversely, inadequate KE with local communities can lead to policy failure, as local knowledge and interests are not heeded and conflicts and distrust can emerge [27–31]. To date, KE has been considered mostly from the perspectives of knowledge creators or disseminators, such as scientists, managers, and policymakers, rather than adequately taking into account the views of other knowledge holders who have a stake in the KE process. This means KE, especially KE that aims to share knowledge, can result in a mismatch between the demand for and supply of knowledge and compromise the effectiveness and efficiency of environmental management [32–36].

In China, meaningful institutional frameworks to engage the public and exchange knowledge in environmental management do not yet exist [11, 37, 38]. Environmental management mostly follows action-oriented, command-and-control approaches [39]. Since 1999, China's National Environmental Programmes (NEPs) have been formulated and implemented to address national land-system sustainability [40]. The Three North Shelterbelt Programme (TNSP), the Grain for Green Programme (GGP), and the Beijing-Tianjing Sandstorm Sources Control Programme (BTSSCP) were designed to combat desertification and land degradation

that had continued in north-western China for decades [41]. Substantial human and financial resources have been mobilised to implement these programmes and compensate local farmers and herders. The government has invested more than ¥500 billion (>$72 billion, $1 = ¥7) in the GGP alone which covers around 35,000,000 ha and has involved 41 million households over the 20-year implementation period [42]. The central government provided the schemes and money and had designated ministries to administer the implementation. Implementation began at the provincial level. Knowledge and information were transferred from higher level agencies of provinces and municipalities, to lower levels such as counties and towns, through meetings and training in the hierarchical administrative system. On the ground, grassroots implementers (street-level bureaucrats) and the staff of local agencies interacted with farmers and herders face-to-face. In contrast to the well-established channels for KE in the bureaucratic system, pathways to enable KE with local farmers and herders were not clear, while studies regarding them remain scarce.

Twenty years later, positive NEP interventions such as afforestation, rehabilitation of mobile sandy land, and water conservation, have substantially improved the local biophysical environment in terms of vegetation coverage, soil erosion control, and biodiversity conservation [40–41, 43]. But afforestation has also caused damage to nearby crops and brought about substantial costs to rural farmers [31], while unpredictable shifts in NEP implementation have created conflicts and complaints among local communities [44]. The state owned Xinhua news agency (2018) warned that grassroots officials were so desperate to deal with tasks and appraisals from superior agencies, they would do no more than the required tasks to avoid the risk of being seen as unconforming [45]. This has dampened the prospects for KE with local communities and increased the possibility of local conflicts.

We seek to address the gaps in research on KE by generating new evidence to inform better KE pathways in China. We investigate KE among scientists, grassroots implementers, and local farmers and herders during the implementation of NEPs focused on desertification and land degradation, aiming to answer the following questions: 1) What knowledge has been exchanged among actors and how? 2) What impacts has KE delivered? 3) What do the actors think of the KE, especially the roles of other actors in the KE process? While acknowledging the complexities the whole situation in China could possess, we seek to improve understanding of KE at local levels and under different institutions through this lens and at the same time, inspire more research to support the creation of more flexible and adaptive strategies for KE in environmental management in the future.

The remainder of the paper is structured as follows. We outline our methodology in the subsequent section. The results section is broken down according to the KE among different actors during the policy implementation process, capturing their respective perceptions and the degree of match to their knowledge needs. The discussion and conclusion sections consider what the findings mean for KE research more generally, as well as for future environmental management in China.

## Methodology

### Study sites

When we conducted our data collection, there were some topics local communities considered to be sensitive. To secure the privacy of the participants, the study sites and participants are anonymised. The study sites consist of three national desertification research stations and the communities surrounding them, anonymised here as A, B and C. We divided the participants into 3 groups as Case 1, Case 2, and Case 3. In the analysis, each participant was designated 3 numbers. The first number indicates which case he/she is from. The 2nd number denotes their

**Table 1. Characteristics of the study sites[1] and the NEPs.**

| Study sites | Dominant ecosystem type | Resident type | Major land use | Major NEPs in place |
|---|---|---|---|---|
| A | Grassland-Forest | Farmers | Rain-fed agriculture | GGP |
| B | Desert-Grassland | Farmers/ Herders | Irrigation agriculture | TNSP/ GGP |
| C | Grassland-Sand land | Farmers/ Herders | Irrigation agriculture-grazing | TNSP/ BTSSCP |

[1], Sources: http://dga.ib.cas.cn/

stakeholder group (1 stands for scientist, 2 for grassroots implementer, and 3 for farmers/ herders). The last number shows the participant's order in the recorded interview or survey in a case. For example, C12-3 represents a grassroots implementer from Case 1 who gave the 3rd recorded interview.

The research stations are located in north-western China where intensive human activities by farmers and herders have made the land vulnerable to desertification and land degradation [46]. All research stations are in the region where NEPs have been implemented (Table 1).

Since their establishment in 1983 (research station in A), 1991 (B), and 1973 (C), several desertification related research, monitoring, and demonstration projects have been undertaken at these stations, generating new knowledge about local social and environmental settings [47]. As members of the Chinese Ecosystem Research Network (CERN), the research stations have received substantial funding from government for infrastructure development and attracted top scientists to conduct their research. Many scientists have been actively engaged in national environmental policy processes [46].

## Methods

We conducted semi-structured interviews with scientists (S1 Appendix) and grassroots implementers (S2 Appendix) based on theme-based, open-ended and/ or multiple-choice questions [48]. A theme-based questionnaire survey was also undertaken to collect household information and views of local farmers and herders on NEP implementation and KE in particular (S3 Appendix).

Ethical approval was sought and granted before the fieldwork began. A Privacy Notice was sent to gatekeeper scientists beforehand, and it was verbally explained in Chinese for each grassroots implementer who could not read English well. Verbal consent was obtained prior to the interviews and surveys. Permission to record the conversations was sought and granted in most cases. One interviewee felt uncomfortable with recording, so notes were taken instead with their agreement. At the end of each conversation, participants were invited to ask any questions to the interviewer, to help the participants better understand the research. Detailed answers were given. These interactions were also recorded when possible or noted down as memos. The unplanned "interviewer question time" often led to further and unexpected conversations and enriched the content of initial interview topics and questionnaires. Due to local Covid-19 restrictions and personal preferences, one scientist was interviewed over WeChat (a Chinese version of "WhatsApp", a messaging application available on mobile phones and laptops), and four scientists were interviewed over email. Both the questionnaire and interview instruments were developed in English and then translated into Chinese. As some farmers and herders were illiterate, questionnaires were administered verbally, which sometimes led to unexpected conversations about which fieldnotes were taken. Most herders spoke Mongolian, for which we worked with a local translator. No minors (people under the age of full legal responsibility) were involved in the research.

We interviewed scientists who had worked at the research stations for over 3 years to ensure that interviewees were experienced in engaging with local farmers and herders. Before face-to-face interviews, we undertook pilot interviews over WeChat with gatekeeper scientists (station directors). They gave feedback on our interview protocol, and introduced us to scientists with different expertise, work experience, and gender for the interviews. In addition, staff and research information were reviewed on the research station websites prior to the field data collection. During the interview, we would ask the gatekeeper scientists to introduce scientists who were potentially representative but had been missed. This allowed us to start from multiple points which helped to avoid the potential linearity associated with snowball sampling. Feedback and introductions from the gatekeeper scientists enhanced trust between interviewees and interviewer and helped secure rich information.

Scientists also introduced us to the heads of local agencies, whose permission was required to access grassroots implementers. It was nevertheless difficult to interview the implementers. Some heads hesitated to introduce their subordinates to researchers from a foreign university. Implementers who had engaged in the implementation of one of the NEPs for more than a year were targeted as they would have interacted with local farmers and herders. Ultimately, those we interviewed had at least 5 years of work experience with one of the NEPs, and some had been working with the NEPs for 20 years.

We obtained consent to interview two heads of the local agencies about implementation process. We considered these agencies as "grassroots implementers" in the analysis. Grassroots implementers and scientists helped us identify household survey participants, guiding us to villages in which one of the NEPs had been implemented. However, they did not accompany us to the villages which helped us to maintain our independence. We sampled farmers and herders who had lived in these villages and been affected by the NEPs for at least 1 year. Convenience sampling was followed in the villages, whereby we knocked on doors and talked with anyone who would like to open them and carried out the survey when consent was obtained [49]. We visited more than 200 households and 187 participants completed the survey, among which, 64 were sampled from 15 villages in study site A, 66 from 6 villages in B, and 57 from 11 villages in C (Table 2). All the villages randomly scattered around the stations and the total population were not calculated as most villages were partially resided and participants had different estimations. In total, 22 scientists from the research stations and 14 local grassroots implementers were interviewed.

Analysis began in the field. Interesting observations were written down as analytical memos at the end of each day [50]. When similar patterns recurred, categories were developed.

**Table 2. Summary of interviews and questionnaires.**

| Actor category | No. and type of date generating meeting/ survey | No. and type of participants |
|---|---|---|
| **Scientist** | 15 interviews | 22 scientists, with expertise in climate change, desertification monitoring, dryland science, pastureland science, plant physiology, small watershed management, sustainable agriculture, sustainability, and water and soil conservation |
| | 1 interview (WeChat) | |
| | 2 pilot interviews (WeChat) | |
| | 4 structured interviews (emails) | |
| **Grassroots implementer** | 14 interviews | 12 grassroots implementers, working on the NEPs from 5–20 years, 2 heads of local agencies working in the position for 3 and 20 years respectively |
| **Local farmers/ herders** | 187 questionnaires | 158 farmers (including contracted outsider farmers[1], village heads, most of Han ethnicity); 29 herders of Mongolia ethnicity |

[1]Contracted outsider farmers were not local people but had moved in from outside and sought contracts to cultivate local lands, as many local people had stopped tilling their fields after migrating to towns and cities.

"Policy" was one such category. We often heard people complain about policy changes such as lower compensation, and people frequently chose "local governments" when they needed help. Other categories such as "institutions", "market", "change", "governance" emerged the same way. Some of the 27 analytical memos were created about the same categories, but with different details as they were recorded daily.

Data from hard-copy questionnaires was digitised after data collection to enable coding and statistical analysis, followed by transcription of recorded interviews. The "Dictate" function in Microsoft 365 (Word) was used to transcribe the conversations. While transcribing, key sentences and details of each conversation in relation to specific questions were manually identified. The holistic coding began at the same time, to address similar patterns and form categories which often overlapped with content of the analytical memos. Key sentences and details of each conversation or answer were translated into English and grouped based on the questions asked. Each conversation in the groups was then summarised, and we searched for patterns, categories, and themes.

While we used thematic analysis as described above, an actor-oriented strategy was taken to present the results [23]. The interfaces between scientists, grassroots implementers, and farmers and herders were observed, alongside opinions from different actors about each other's knowledge. Preferred methods of KE were identified, along with knowledge needs, providing a more open way of looking at interventions and the interlocking of arenas pertinent to KE in the implementation process [23]. Descriptive statistical analysis was conducted with Excel (version 16.66.1) on the questionnaire data, to discern trends and patterns in socioeconomic characteristics of local farmers and herders.

## Results

The results are presented in 3 sections:1) KE during the implementation of NEPs; 2) the impacts of the KE; and 3) perspectives about KE among the actors. Considering the complexities during the KE, section 1 is further broken into 2 sub-sections: KE with policymakers, and KE among scientists, grassroots implementers, and farmers and herders, and followed with a brief summary. To clearly demonstrate the complicated interaction among the actors, each interface is positioned as the 4th level heading with numbering before it.

### KE during the implementation of NEPs

**KE with policymakers.** *1. Scientists and policymakers.* Most scientists considered that they had actively engaged in the formulation of the NEPs, with some reporting direct involvement. When asked how they got involved, most scientists mentioned surveys and online questionnaires from governments and local agencies. Some senior scientists were regularly invited to lecture at training sessions for officials of local environmental agencies; others were members of NEP Assessment Teams or other environmental programmes; and others became members of the taskforce to set up a NEP. Perspectives nevertheless differed on how they engaged: "*I used to receive questionnaire letters from various governmental agencies. There are more online now. We are surely engaged*" (C31-2). Another noted: "*We were in a consulting role. We gave suggestions. But whether they listen to us or not, we don't know*" (C31-3). Senior scientists who work for national and provincial research institutes felt more strongly that they had engaged with policymakers, which contrasted with the responses of junior and local frontline scientists.

All scientists felt that progress had been made in KE with policymakers. One scientist shared an example: "*Most of us are reading Ecological Conservation and High-quality Development Guidelines for Yellow River Watershed which was issued by the State Department. You can*

*tell it must have been prepared by scientists. The language, terms, and concepts are academic and professional, and of the cutting-edge*" (C31-2).

Further insights emerged from scientists' involvement in implementation. A frontline scientist, who had worked in desertification rehabilitation and prevention for over 20 years, reflected: "*The Three North Shelterbelt Programme (TNSP) covered several provinces and they* [the governments] *decided which municipalities and counties under their administration should enter the Programme. When our county was chosen, it was the local forestry agency that was in charge of the implementation. Did scientists participate? A very small group. Have the scientific findings been used? Definitely yes. Were they fully used? I don't think so*" (C11-3). A senior scientist corroborated this from another angle: "*Our NEPs were often entrusted to planning and designing institutes for details and materialization once concepts and decisions had been made by policymakers and scientists. Staff of the institutes knew very well about the planning and design procedures. They would not make scientific mistakes either. But compared with scientists who are always active in research, their knowledge usually has not been updated in a timely way. When they worked out the plan, many things had already changed*" (C11-8).

Despite limited involvement in planning, design and implementation, scientists were keen to work with policymakers to address environmental issues despite their different approaches (e.g., theoretical vs. applied, local vs. regional). Governments and policymakers were their first or second choice when asked "with whom would you like to share your knowledge".

*2. Grassroots implementers and policymakers.* All grassroots implementers could receive regular training from policymakers, which was the most important KE opportunity for them: "*Training had different levels. There was training for grassroots implementers, and for staff and officials of higher administrative levels, e.g., of county and municipality. Practical skills were the main content of grassroot implementers' training. For those at higher levels, policy details were the focus. Their training would also show them how to design plans to have their subordinates trained*" (C12-3). The above recollection resonated with another interviewee who was the head of a county environmental agency: "*I attended training meetings convened by the Autonomous government and the municipal government. Coming back, I convened my subordinates and explained relevant policies and shared with them various operation manuals I received at the training*" (C12-1). All grassroots interviewees agreed they had received the necessary support and resources to complete implementation tasks.

Not all grassroots implementers received the same training, but they learned from each other when needed: "*It was rather daunting to walk to measure the area of thousands of hectares* [of retired lands] *with a handheld GPS. Later the municipal government equipped us with a drone. Two of our colleagues were then sent to learn the operating skills and how to analyse with ArcGIS. They now lead us to use the skills*" (C22-6).

Grassroot implementer attitudes toward training were positive but varied. Some considered training indispensable: "*Training is not only about learning skills. It is also about getting familiar with policies. . .training is very necessary for grassroots implementers who are at the frontline with rich experience. They often believe in their own experience more. . .Through the training, they have an opportunity to access new skills and ideas*" (C12-3). Others hoped for more training beyond the programmes in areas such as "*some systematic theories and management skills about agroforestry*" (C22-1).

Usually, grassroots implementers would follow the guidelines from their training and try to complete their assigned tasks given the strict criteria for post-implementation assessment. Reflective communications from them to their superiors or policymakers were rare. However, when strong pushbacks from local farmers or herders happened, even village heads who were often allies of implementers, supported the local people. Implementers would then report unsurmountable "barriers" to their superiors to seek changes in involved policies, such as a

**Table 3. Age range distribution among surveyed farmers and herders.**

| | | Age range | | | | Total |
|---|---|---|---|---|---|---|
| | | 18–30 | 31–50 | 51–65 | 65+ | |
| Farmer Herder | | 1 | 40 | 78 | 39 | 158 |
| | | 4 | 20 | 4 | 1 | 29 |
| Total Percentage of Total | | 5 | 60 | 82 | 40 | 187 |
| | | 2.7% | 32.1% | 43.9% | 21.4% | 100.0% |

ban on use of established forests. In this case, the implementers, based on their experience, testified that "*moderate grazing of the forest floor can reduce fire risks*" (C22-5), which later led to changes in local forest management policy.

*3. Local farmers/ herders and policymakers.* Of the 187 participants, over 20% were aged 66 or above, about 32% in their 30s and 40s, almost 43.9% aged 50–65 years old, and only 2.7% were under 30. There were more younger herders in the survey (Table 3). Most young farmers, especially those in their 30s or younger, had moved to towns and cities for job opportunities and better earning potential. It became increasingly difficult for farmers to find even temporary jobs from their late 40s, and they had to stay on the land. Age was also an issue for herders to migrate to towns and cities for jobs. Only a few young herders managed to do so as they had inherited a relatively large area of pastureland for livestock breeding. More details on this will be provided in the following sections.

When asked how they received daily information about the outside world, the pathways to receive and communicate information and skills were different among different age ranges (Table 4). Most farmers aged 66 and over mentioned TV. For those who were below 50, mobile phones were key for obtaining information, including on agricultural products and treatment of livestock diseases. Of the 40 investigated, 36 of them chose social media as their major pathway to access and exchange information. Some aged 50–65 preferred TV, some mobile phone,

**Table 4. How farmers and herders receive and communicate information.**

| | | | Social media | TV | Talking with friends | Other |
|---|---|---|---|---|---|---|
| Farmer | Age range | 18–30 | 1 | 0 | 0 | 0 |
| | | 31–50 | 36 | 12 | 2 | 5 |
| | | 51–65 | 61 | 45 | 7 | 6 |
| | | 65+ | 20 | 33 | 2 | 4 |
| | Total Percentage of Total | | 118 | 90 | 11 | 15 |
| | | | 74.7% | 57.0% | 7.0% | 9.5% |
| Herder | Age range | 18–30 | 4 | 0 | 1 | 2 |
| | | 31–50 | 19 | 3 | 2 | 7 |
| | | 51–65 | 3 | 1 | 0 | 1 |
| | | 65+ | 1 | 0 | 0 | 0 |
| | Total Percentage of Total | | 27 | 4 | 3 | 10 |
| | | | 93.1% | 13.8% | 10.3% | 34.5% |
| Total | Age range | 18–30 | 5 | 0 | 1 | 2 |
| | | 31–50 | 55 | 15 | 4 | 12 |
| | | 51–65 | 64 | 46 | 7 | 7 |
| | | 65+ | 21 | 33 | 2 | 4 |
| | Total Percentage of Total | | 145 | 94 | 14 | 25 |
| | | | 77.5% | 50.3% | 7.5% | 13.4% |

while others used both. While the older farmers passively receive information from TV, younger farmers use mobile phones proactively to receive and share knowledge, using and enhancing their agency. Social media was chosen by 118 out of the 158 farmers.

The pattern was similar with herders. 27 of the 29 participants chose social media, and only 4 chose TV in contrast to 90 for farmers. One of the reasons for this is that herders were of Mongolian ethnicity and senior herders of 66 years old or above did not speak or understand Mandarin. The 50–65-year-old herders used Mandarin for basic communication. When asked how they accessed information about policies, many of them mentioned their village heads who were also Mongolian. Younger (below 50 years old) and well-educated herders could speak Mandarin well and used it as another daily language when speaking with people of Han ethnicity.

Sharing information and communicating with others has become easier thanks to technology. Almost 90% of the surveyed farmers and herders had mobile phones. Apps on their phone provide platforms for sharing their experiences, seeking help and identifying opportunities. "*We have various apps and chat groups on the phone. When we need to say something, we just text it or voice it, then press 'send'*" (C13-25).

However, farmers said that they did not get all the information they needed about the NEPs from policymakers or governments. An example of this was a new policy for the second stage of the GGP which reduced compensation rates [42]. Most farmers were unaware of the changes and no official explanation about the changes and reasons for them had reached them. One complained: "*The cash compensation at the beginning could buy several bags of rice and white flour, supporting my family quite well.* [Now] *it can only buy a bowl of noodles*?! *No help at all. The village committee might embezzle our compensation without our knowing*" (C13-33). Lack of timely KE led to doubts and distrust fermenting among local communities.

**Knowledge exchange among scientists, grassroots implementers, farmers and herders.** *1. Scientists and grassroots implementers.* Scientists had a positive view on grassroots implementers' knowledge about dealing with desertification and land degradation: "*they have been working hard*" (C31-2), and "*they know very well about local environmental, socioeconomic situations*" (C21-1 and C21-5). Some scientists explained how they helped grassroots implementers with new planting and monitoring skills, but most of them considered that there was not much KE because the links between their research and activities of grassroots implementers were weak.

Scientists at the research stations had worked in the past side-by-side with local governments and grassroots implementers. One senior scientist mentioned that "*when there was no precipitation for a month, we would investigate the impacts and work out solutions which would then be printed in local official bulletins and broadcast to the whole county. . . .and we were always invited when the county government convened meetings for local agricultural production*" (C31-5). However, grassroots implementers in county environmental agencies were in charge of NEP implementation. In a hierarchical administrative structure, they received training from agencies above, such as the municipal environmental agencies, and needed to meet their demands and report to them. KE with scientists did not happen much according to grassroots implementers. However, all implementers indicated that they had communicated with scientists during implementation, although only a few could specify the interactions. When grassroots implementers were asked what kind of knowledge they needed, most mentioned support with practical issues, such as how to ensure survival of planted trees, or how to maintain the forests. "*The scientists should descend on the ground and help us*" (C12-2), an implementer said, referring to the situation whereby large-scale and hot topics were more likely be funded by the government and on scientists' research lists.

*2. Grassroots implementers and local farmers/ herders*. KE between grassroots implementers and farmers and herders mostly took place at the start of implementation when policies were clarified so that farmers would accept the measures to be implemented on their land. Grassroots implementers needed to contact heads of village committees first to convene village meetings to explain implementation measures, and address farmers and herders' concerns about compensation, ownership, and other support from NEPs.

After mobilisation and policy clarification meetings, "*we then wait for applicants* [farmers/ herders who voluntarily accept measures from the NEP]. *We cannot force them. Only when they told us where and what size of the land to be retired for trees and grasses could we begin to investigate their lands, measure the size and geo-reference them on the map. After that, we would allocate seedlings to them for free, providing technical support if they need*" (C22-6), recalled a grassroots implementer. Another implementer mentioned a different scenario after the mobilisation meetings: "*We distributed our afforestation task* [aiming to accomplish a certain forested area in a certain year] *to individual villages. When a task was assigned to a village, we needed to identify where the lands lie, calculate the areas on the map...The area of land was directly related to the compensation a farmer could receive- we were very careful in this regard. Then, we organised them to plant trees, showing them key technical requirements in preparing land and planting*" (C32-2). Even in the same NEP, the implementation process and KE between local farmers and herders and grassroots implementers could differ among various administrative regions.

Once planting of trees and shrubs began, some farmers were employed by the grassroots implementers to help. During this period, KE focused on technical issues and awareness raising. "*We taught them* [the farmers] *how to plant trees in sand lands, and told them when the trees grew up, the environment would get better*" (C12-2). Grassroots implementers believed that the experience with the NEPs not only improved their environmental awareness, but also that of the engaged farmers and herders.

However, only a few farmers or herders chose "grassroots implementers" when asked "whom they would turn to for help". We heard many complaints about reduced compensation, changed property rights over the established forests, and bans preventing them from using forests established on their lands. During the survey, many farmers and herders considered that implementers kept close watch over the forests and prevented their use, without providing adequate compensation. Grassroots implementers stuck to their instructions from above, considering they were protecting the environment for the common good. The inconsistency of policies and the lack of effective communication about changes in them spiked distrust among farmers, herders, and grassroots implementers.

*3. Farmers/ herders and scientists*. When asked whether farmers and herders had knowledge to deal with desertification and land degradation, all the scientists agreed. Three of them even believed that farmers' solutions to local environmental problems were sometimes simpler and better than those promoted by scientists. One said that "*they* [the farmers] *may be poor. They may not read many books, but they are not stupid. They know their lands well*" (C31-5).

While all scientists met farmers or herders during their field surveys, only two were in regular contact with them as part of their current research projects. Many scientists mentioned that they had experience with demonstration projects, but they also admitted that the projects mostly focused on promoting new tree or shrub varieties that adapt to local arid and semi-arid conditions, or showcasing techniques for restoration of local grasslands, which were quite different from those before 2000 when efforts had focused on farmers' and herders' need for food. Scientists' active engagement in demonstration projects to help local communities deal with desertification and land degradation has gradually changed. Most said they were working in specific disciplinary fields which did not need interactions with farmers or herders. Besides,

*"they* [farmers] *used to focus on producing more food from the degraded fields which scientists were able to help. But they now are eager to grow market-successful products. That is beyond the scope of scientists' knowledge"* (C11-7).

About one fifth of the farmers and herders (22%) said that they had attended demonstration projects facilitated by scientists and local governments, and all attendees considered the projects "helpful" or "very helpful". They had been shown how to manage orchards, grow vegetables in greenhouses, cultivate drought-resistant seeds, and raise new breeds of sheep adapted to confined breeding. However, all farmers and herders said that they "seldom" met scientists in recent years, which was corroborated in interviews with scientists. Although some of them wanted to learn from scientists how to improve soil quality, choose appropriate fertilizers, or tackle crop pests, when asked to whom they would turn for help, few farmers and herders chose "scientists", but rather "local governments" and/ or "village heads". Farmers and herders believed that policies rather than knowledge were shaping their livelihoods.

To summarise, KE occurred between senior scientists and policymakers during NEP formulation. However, junior and frontline scientists hardly participated in policy formulation or implementation. This prevented the inclusion of local contextual considerations into the NEPs, and assessment criteria were sometimes compromised on the ground. Neither scientists nor grassroots implementers had the motivation for, or were supported to undertake, KE during implementation. Although KE featured in the administrative system, it was often one-way (from superiors to subordinates). Grassroots implementers and village heads actively exchanged knowledge with farmers and herders to complete tasks given by their superiors. However, there were no mechanisms to incorporate and communicate farmers' and herders' concerns. Unless intense, large-scale and rare pushbacks erupted, they would stick to the tasks, omitting to reflect on local matters with policymakers or in the NEPs. Technological advances have provided pathways for farmers and herders to gain knowledge from the outside world and exchange knowledge amongst themselves, but they remain largely knowledge receivers and unable to access the knowledge they really need. They also lack the pathways to communicate their knowledge needs to knowledge providers (Fig 1).

For the 3 cases, a synthesis of observations from scientists, grassroots implementers, and farmers/ herders about KE with each other during the implementation processes has been developed (Table 5).

## The impacts of KE from the implementation of the NEPs

**Frontline knowledge cannot be sufficiently addressed by scientists.** Although all participants had observed environmental improvements since the start of the NEPs, grassroots implementers, farmers and herders questioned whether NEPs had adequately addressed the situation on the ground. Some ***implementers*** doubted the NEP criteria could be fully met locally since "*they* [scientists] *did not take local conditions into consideration at the beginning*" (C12-2). When asked what kind of knowledge they thought scientists needed, some implementers believed: "*scientists know the bigger picture very well, but often miss local specific details*" (C22-1). The situation seems to have little chance of improving as the survey revealed the absence of engagement of scientists among local communities.

Frequent exchanges with policymakers during NEP formulation were reported by senior scientists, which helped to secure effective and efficient responses to national environmental challenges. However, regional and local problems have not been fully addressed. Institutional support for junior and frontline scientists to participate in formulating or implementing processes was limited. Frontline knowledge, i.e., understanding of local situations and practical issues confronted on the ground in the changing social and biophysical environments during

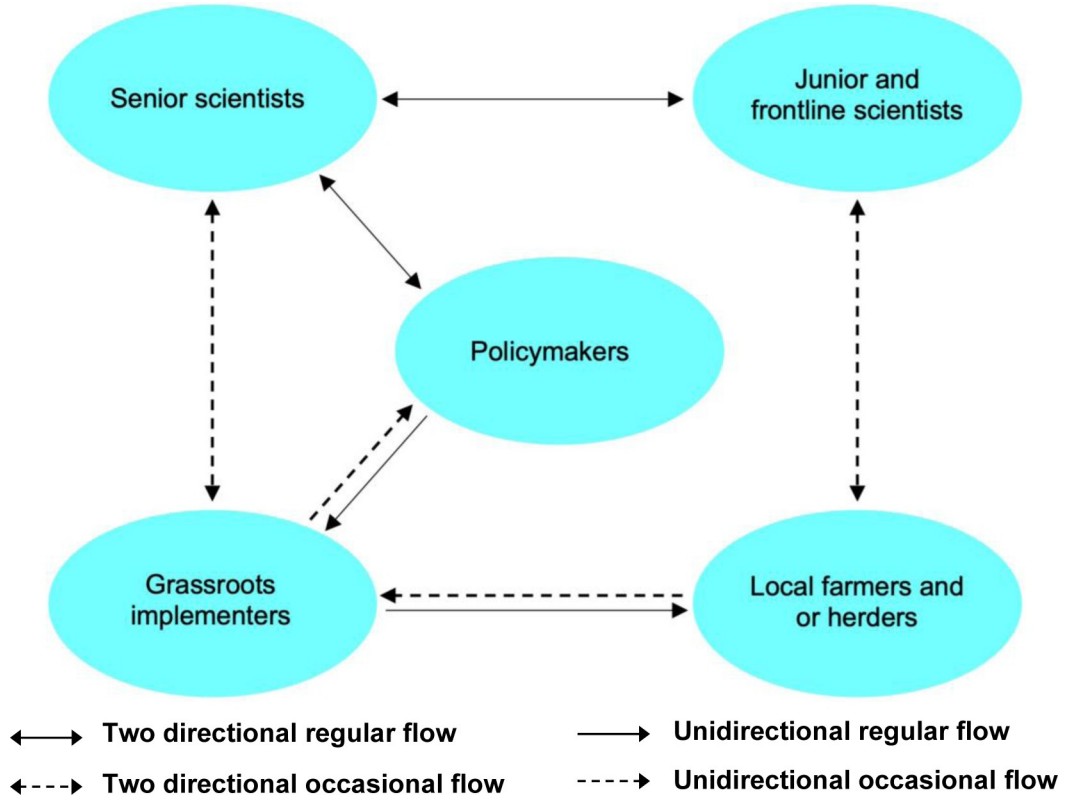

**Fig 1. Knowledge exchange during the implementation of national environmental programmes.**

implementation, has not been generated, collected, or exchanged satisfactorily. Moreover, the priorities of NEPs have evolved. GGP, for example, aimed to reverse land degradation and restore the environment in its first stage (1999–2007) while its second stage (2014–2019) targeted poverty eradication and sustainable rural development [42]. New frontline knowledge is needed to appreciate diverse and specific situations and achieve such evolving goals.

**Table 5. Summarised observations from scientists, grassroots implementers, and farmers/ herders about KE with each other in the 3 cases.**

|  | Case 1 | Case 2 | Case 3 |
|---|---|---|---|
| **Scientists** | • Grassroots implementers work very hard.<br>• Farmers are poor and don't read many books, but they are not stupid. Some of their knowledge works better than solutions from scientists in the field. | • Grassroots implementers have rich practical experience.<br>• Scientists have learnt a lot from farmers and herders. | • Grassroots implementers know best about local socioeconomic situations as well as limiting factors that could affect policy implementation.<br>• Grassroots implementers and local farmers know better which approaches are more effective in tackling desertification in their communities. |
| **Grassroots implementers** | • Implementers learn the latest trends from scientists. Some of their research could be more practical, adapting to specific, real-life situations.<br>• Farmers need to learn skills and find ways to maintain the standard of livelihoods. | • Scientists are experts, but their research should descend to the ground and focus on practical issues. Farmers have no knowledge. Farmers need support from the governments. | • Scientists know very well about the overall situation and general trends but they sometimes miss some details about the local environment and communities.<br>• Farmers and herders are well educated and know their business very well, but they need support from governments about market information for agricultural production and products. |
| **Farmers/ Herders** | • Scientists are not helpful in solving issues in the fields.<br>• . . . (do not want to talk about grassroots implementers.) | • Scientists are experts, but farmers cannot understand them.<br>• . . . (do not want to talk about grassroots implementers.) | • Science is important for farmers/ herders.<br>• Farmers/ herders get various information from village heads (who have close connections with grassroots implementers). |

**New doubts and distrust have emerged.**   Doubts and distrust manifested especially between grassroots implementers and local farmers and herders. Some implementers commented that farmers and herders were "educated", "understanding" and "cooperative" while others described them as "selfish", "ignorant" and "short-sighted". Behind each positive comment there was always a positive relationship among implementers, farmers, and village heads that demonstrated good communication and trust during the implementation of the NEPs.

Survey results indicated how planting trees and shrubs on farmers' or herders' land, and then enforcing regulations to prevent their use, had led to conflicts between implementers and land users. In some areas, raising sheep had been a pillar of local livelihoods and major income source for most families. Under NEP implementation, grazing was banned. Despite encouragement for confined sheep raising, there was insufficient land for forage production which forced most farmers to change their way of making a living. Of the 64 families around one of the research stations, 51 possessed less than 1 ha of arable land. The reality was, except for the herders who could access larger pastureland sometimes of more than 100 ha, over 47% of land users had less than 1 ha of arable land at their disposal. While land is a precious asset for farmers and herders, over 41% of farmers lost half of their land under the NEPs.

Farmers and herders frequently complained about compensation and property rights over their retired land. Official documents capture policy changes but information about the changes had not reached all local land users. Reports about grassroots implementers and village heads who hid policy information from farmers and herders and embezzled or appropriated compensation funds have emerged [51]. The dynamics of the NEPs, compounded by poor communication from grassroots implementers to their superiors, and farmers' and herders' inability to communicate with governments and policymakers, have created new tensions among local communities.

**Local farmers'/ herders' concerns cannot be sufficiently addressed by NEPs.**   Farmers were amused when being asked "whether they have enough food" and assured us food was no longer a problem. Electricity was their main energy source, although more trees were around and collecting fuelwood had become easier thanks to the NEPs. Transportation also improved considerably over the past 20 years. Well-established road networks connected most households in all villages. Indeed, not only has the biophysical environment been enhanced; substantial socio-economic changes have happened as well.

When asked whether they have any worries now or regarding the future, 48% of respondents mentioned "lack of affordable social care services"; 70% worried about rising prices of fertilizers, seeds, and agricultural machinery; and 25% were upset about declining groundwater and deteriorating soil quality. As a scientist described "…*the rural villages act like a giant pump. They are pumping precious water from the rural areas to moisturise towns and cities. They have no idea how long it can be sustained*" (C11-6). As supportive policies were lacking, land users tried to make the most out of the limited land resources to cover costs of agricultural production and save money for future social care, both of which would be bought from towns and cities.

As social policies take time to be implemented, policymakers and governments could do more in terms of KE to support farmers and herders. Both scientists and grassroots implementers mentioned farmers were often too ready to jump on a bandwagon to produce market goods, which often caused supply to exceed demand and led to substantial losses. One grassroots implementer noted: "…*. they* [farmers] *lacked information about other planters. They had no knowledge about the market. They only believed successful stories and hoped to become one of them. But the market changes fast and there is a demand ceiling. When everyone tries to replicate the story, the market cannot absorb the flood in supply*" (C12-4). Many participants suggested governments should have the capability to collect information on supply and demand, but that they failed to organise and share it.

In summary, KE among senior scientists and policymakers secured the effectiveness and efficiency of NEPs in terms of their initial goals of solving environmental problems. However, inadequate participation of junior and frontline scientists has impaired the NEPs' abilities to solve local and practical issues during implementation. Tensions between grassroots implementers and local farmers and herders were partly caused by insufficient KE between policymakers and affected farmers and herders, and partly due to the evolution of the NEPs. Farmers' and herders' concerns that relevant policies or more KE might help, were overlooked, resulting in socio-economically unsustainable outcomes from the NEPs (Table 5).

## Perspectives about KE with each other

**Scientists.** In interviews, all scientists chose "academic settings (journals, conferences, workshops)" as the most common pathway to communicate research findings, with some highlighting that publishing papers was their priority for career reasons. While some scientists acknowledged communication with local communities was not enough, they reasoned that they lacked due support: "*We* [frontline scientists] *cannot get enough* [financial] *support ourselves. We have to join 'big' scientists and follow their plans and perceptions*" (C11-6). One of the frontline scientists said: "…*we regularly submit the monitoring data we collected from the field station to the* [national network monitoring] *system, and some researchers in the office will use the data to produce papers. They then get promoted. We work with the local meteorological agency and health agency and use the data to alert the farmers and local community of extreme weather conditions or poor air quality, but still we are required to compete with those scientists in paper numbers for limited funding and promotion chances…*" (C11-5).

His account highlighted the role of the current assessment system. The number of peer reviewed papers scientists have published affects their career from funding opportunities to promotions and social influence. A senior scientist recalled that "…[agricultural] *field experiments needed at least 3 years to deliver complete and reliable conclusions in the past. But fewer and fewer scientists would spend that time in that harsh environment for such limited outcomes* [i.e., lack of opportunities for papers and influence]" (C31-4).

**Grassroots implementers.** Communications with local farmers and herders were considered necessary, especially as trees and grasses were planted on their land and their help was often needed for planting. For implementers, the priority was to complete planting tasks and ensure they met the assessment criteria of the superior agencies. "…*We hope to share our knowledge and experience* [in planting trees] *with them. We'll try every possible way to implement the GGP….*" (C32-1). Besides,"*The policy was good. If it was clearly explained, it was not difficult to communicate with them* [farmers]. *Most of them were reasonable*" (C32-2). These examples represented grassroots implementers' opinions about KE during implementation. Implementers who only met farmers in spring for afforestation were more negative about farmers' environmental knowledge and awareness, compared with those who frequently visited farmers and had more positive attitudes. Some implementers admitted their families, friends, and relatives were among the farmers and herders, and they understood them and would support them with information and skills. Yet, when forests became established, conflicts with neighbouring farmers and herders emerged. "*We need to constantly tell them not to let goats or sheep into the forests. Sometimes, we would let them in to collect fallen trees or forage for their livestock to make a temporary peace*" (C22-6).

Implementers' perspectives on KE with scientists kept changing as implementation progressed. When in the planting stage, they complained scientists should have been in the field before they developed criteria in their offices; many hoped scientists would help them with practical issues in the maintenance stage, such as treatment of tree pests and diseases, forest

**Table 6. Farmers and herders' choices to "whom they would turn to for help".**

| | Central government | Grassroots implementers | Local governments | Scientists | Village head | No choice made | Other |
|---|---|---|---|---|---|---|---|
| **Farmer Herder** | 2 | 13 | 44 | 12 | 51 | 42 | 9 |
| | 0 | 2 | 6 | 3 | 13 | 5 | 3 |
| **Total** | 2 | 15 | 50 | 15 | 64 | 47 | 12 |
| **Percentage of Total** | 1.1% | 8.0% | 26.7% | 8.0% | 34.2% | 25.1% | 6.4% |

diversity, and timber harvest and utilisation. Considering the roles scientists played in the NEPs, grassroots implementers requested more support and KE with frontline scientists rather than interaction with senior scientists.

**Local farmers/ herders.** When asked to whom they would turn to for help, almost half of the herders chose "village heads" and one third of farmers would do the same (Table 6). A total of 28% of farmers and 21% of herders chose "local governments". Although grassroots implementers had actively engaged with them, only 8% of farmers and less than 7% of herders considered approaching them for help. "Scientists" were also not often asked for help, just slightly more popular among herders than farmers. Interestingly, 25.1% of participants chose no one as a source of help. Fig 2 indicates some of the reasons for these choices.

Those who chose "village head" did so because village head was "often being available". "Local governments" were considered to be able to offer "more accurate and reliable" information. Lack of trust was a common reason not to choose to ask for help.

The discussions around the survey indicated that herders were concerned about environmental changes in their pasturelands. Many talked about extended droughts and decline and pollution of groundwater, for which they believed the energy companies who were supported by local governments were responsible. Compared with village heads who help them translate policy information from Mandarin to Mongolian, and scientists who study environmental

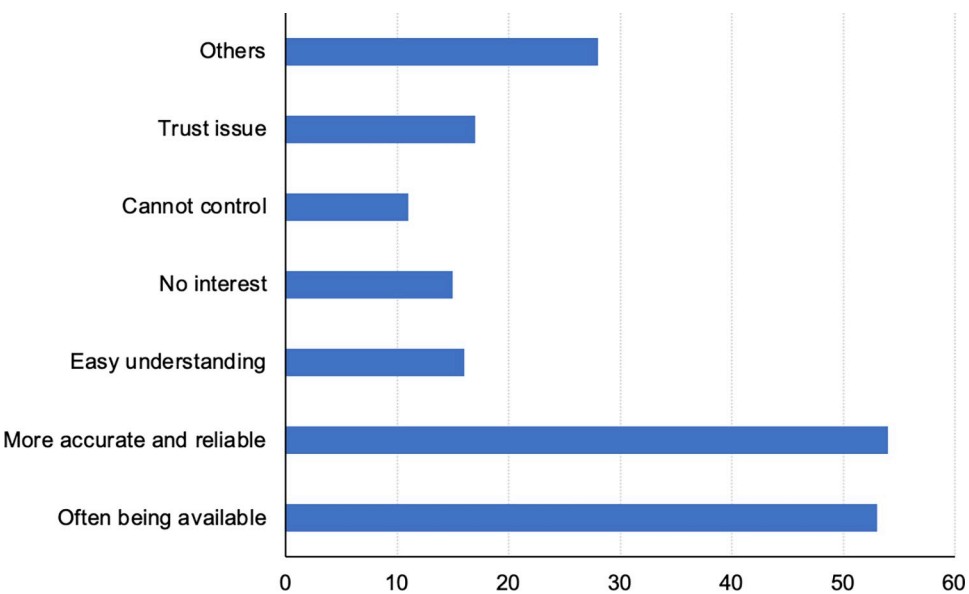

**Fig 2. Reasons why the farmers and herders made the choices of "whom they would turn to for help".** X axis: the number of people who chose the reason Y axis: the reasons behind the choices to the question.

issues, local governments, including grassroots implementers who are official staff, were not popular.

Farmers and herders engaged different ways of living and their KE experiences were also different from the herders'. Almost all farmers who were less than 60 years old had had temporary jobs, such as planting or construction, to support their families. They had chances to engage in social groups from different walks of life. However, few herders had temporary jobs as they were occupied all year round on the pasturelands. Village heads were active in organising processing and contacting potential buyers for herders' produce. During the survey, one village head showed us their village's sheep fleece in a huge storehouse. Village heads played a much more important role for herders than for farmers.

Livelihoods also defined daily priorities, knowledge preferences, and attitudes towards other actors who might affect them. Farmers had access to a much smaller land area than herders. The maximum pastureland area under a herder's control in the survey was approximately 267 ha and the smallest area was about 80 ha. In contrast, the largest farmland holdings were less than 30 ha, and the smallest was about 0.13 ha. Thus, farmers cared more about the short-term economic contribution of the farmland rather than the environment. As forest resources were controlled by grassroots implementers, they did not like the implementers much either. Only 13 farmers chose grassroots implementers when asked whom they would ask for help. Those who chose "local governments" said it was because they did not trust the grassroots implementers and hoped governments could help them.

## Discussion

Our study on KE in the implementation of NEPs in China provides insights into social settings in the developing world where participation of stakeholders is poorly developed and insufficiently weighed in during policymaking; where social welfare systems are not well established, and where financial wellbeing is often prioritised over environmental benefits for local communities. In China, the economy grew, public environmental awareness improved, and government investment in environmental protection increased dramatically [52]. While KE in such contexts is complex, common ground does exist for KE per se.

Matching knowledge supply and demand is prerequisite for effective KE. To reconcile the demand for and supply of science, diverse knowledge users need to be recognised [16, 53]. While not all scientific research directly contributes to environmental management [54], scientific research alone cannot provide all the knowledge that effective environmental management needs, especially the kinds of knowledge needed during implementation. KE among scientists, grassroots implementers, and local farmers and herders has been essential in the implementation of the NEPs, but as the situation evolved, more actors, such as social scientists, entrepreneurs, as well as local governments, were needed to provide different kinds of knowledge considered essential by local communities. Unlike policymakers and grassroots implementers who have specific goals and tangible criteria for the managed environment, local farmers and herders had wider concerns. The environment is just one component of their daily lives. Even in developed economies, engaging actors are advised to acknowledge and take actions to mitigate wider challenging social contexts when adhering to best practices in multi-stakeholder collaboration [55]. André et al. (2021) noted that even when knowledge is co-produced with scientists, practitioners do not always find it actionable as they are also balancing other local needs [17]. Indeed, when implementation of environmental policies touches the ground and begins to intervene in the complex social-ecological systems, many sectors and lifelines are affected [56]. Given the intrinsic unpredictability, nonlinearity, and adaptability of the social-ecological system [57], relevant knowledge production should consider broader

environmental as well as social contexts and the different types of knowledge needed from various actors. Engagement of a wide array of actors is one of the key ingredients in building capacity for environmental management [58, 59]. For scientists thus far, much remains to be learnt as to how science can play a role in multi-actor KE scenarios in environmental management.

Effective KE is also affected by other factors. Motivations are very important for scientists to engage in KE [60], but still there are mechanisms that keep them out of effective conversations, such as cultural differences and institutional barriers [9], or different perceptions about ecological knowledge [3]. First, scientists were not incentivised to engage in KE outside the scientific community. Current assessment systems of scientists are scientific output and award oriented, meaning scientists are less keen on small, local issues that do not attract broad attention [61]. Although junior and frontline scientists (alongside social scientists) could play a big role in helping local communities to identify policy support needs and solve practical problems, they lack financial support and have had few chances to engage [62]. This makes them join senior scientists to undertake funded research on predefined topics that are not in tandem with local knowledge needs. Lack of motivation/chances for on-the-ground research meant that scientists largely failed to share their knowledge with implementers and local farmers and herders. It has also led to lack of on-the-ground research about matters where local contexts failed to match the general assessment criteria formulated by scientists during NEP implementation.

Second, inconsistencies in NEP related policies often cast doubt on implementers' operations in local communities and compromised their autonomy and discretion [22]. When there was no timely communication from local governments to affected local communities, distrust and conflicts arose between implementers and farmers and herders. Heberer (2014) revealed local governments and implementers became too occupied by different tasks and appraisals from superior agencies and gave insufficient attention to matters beyond their immediate tasks [32]. At the same time, increasingly authoritarian measures in environmental management from the central government have forced local governments to focus on environmental protection but has not encouraged them to solve other socio-economic issues occurring in the processes of achieving that protection [63, 64]. All these institutional barriers dampen the chances of KE among local actors.

Third, there is no tangible combined mechanism for monitoring and adapting to the dynamics of socio-ecological systems in environmental governance in China. In the developed world, diverse and well-established research activities around adaptive management have been vigorously explored by scientists. Stewart et al. (2014) even proposed Knowledge Interaction (KI) as supplementary to KE to facilitate interactions with organisational systems and cultures, while some principles of agile management should be applied for adaptive planning, evolutionary development, and continual improvement [65]. Back in China, institutional barriers have kept scientists away from situations on the ground which has been changing constantly. The implementation of NEPs changed the local biophysical environment dramatically in two decades in terms of increased vegetation coverage, fewer sandstorms, and improved crop yields [41]. The role of scientists has shifted from dealing with land degradation and producing enough food, to supporting stable production under a changing climate, forest management, and its sustainable use. Products of farmers and herders are becoming part of global food systems. They need knowledge to retain competitiveness and secure the safety of their own food and local environment.

Hudson et al. (2019) suggested that those who work on the front line, whether managerially or professionally, know more about the challenges of delivery than national policymakers [66]. Formal and informal engagement with practitioners can make scientists more aware of their

evidence needs and can help to speed up the production of relevant and accessible evidence [67]. Current KE deficits in China among scientists, grassroots implementers, and farmers and herders led to knowledge demand from local communities overlooked by scientists and local contexts missed many chances of being reflected into policies. Moreover, conflicts between grassroots implementers and farmers and herders were created mostly by the inadequacy of KE between the latter and local governments who failed to organise, disclose, and share relevant information and policies. Li (2011) observed local governments did not have the pressure to release information to the public until environmental NGOs joined in and developed webs of dialogue [68]. However, environmental NGOs in China are yet to play meaningful and effective roles in advocating environmental protection [69]. Lack of diversity of knowledge and actors in China's environmental management, from a long-term perspective, will ultimately stifle its environmental governance capacity.

## Conclusion

KE is an indispensable part of environmental governance, not only because it is crucial for policymaking, but because it helps solve implementation issues on the ground. This study included grassroots implementers, and local farmers and herders, investigating their knowledge needs and perceptions about existing KE, an area which has received relatively less attention especially when compared with their influence on implementation. We found a significant absence of scientists in the KE during the implementation of NEPs, which meant demand for front-line knowledge could not be met and local scenarios were not well reflected in policies. While grassroots implementers could get enough support from governments through KE, the failure of governments to meet demands for KE from farmers and herders saw development of conflicts between the implementers and local people, while the effects of KE about NEPs were eroded by farmers' and herders' other concerns about living and livelihoods. We conclude that to facilitate successful KE, supply of and demand for knowledge and information should match, for which favourable and supportive institutional arrangements are necessary. Furthermore, complex and dynamic socio-ecological systems require KE to change with changing contexts. Given the emergent and specific demands from grassroots implementers, farmers and herders, effective and efficient KE in environmental management also needs engagement of multiple actors with diverse backgrounds, such as scientists, economists, socialists, entrepreneurs, and NGOs. In the context of China, governments could have played bigger roles in enabling KE among various actors.

Considering the sparse opportunities for engagement in or having influence on decision-making processes in China, multi-actor perspectives are especially significant in informing future national environmental governance. Also, statutes and programmes for environmental governance should become more holistic to help build overall resilience in China's rural areas. This requires more consideration to be given to the root causes of farmers' and herders' behaviours, such as the need for economic returns and social welfare.

Addressing these issues in the context of China provides important insights that are also relevant to other locations in the developing world, despite China's unique system of governance. Besides demonstrating the complexities of knowledge demands on the ground, we also expect to inform future studies of external pressures such as globalization and climate change that have already put farmers and herders in a challenging position: in addition to environmental stewardship, they now need to maintain their yields and find a market for their products. To support them is to invest in food security and social security, as well as environmental sustainability.

## Supporting information

**S1 Appendix. Topics with scientists.**
(PDF)

**S2 Appendix. Topics with grassroot implementers.**
(PDF)

**S3 Appendix. Questionnaire survey.**
(PDF)

## Acknowledgments

We would like to thank the gatekeeper scientists and local assistants. We also thank our anonymous reviewers for their constructive and helpful comments.

## Author Contributions

**Conceptualization:** Zheng-Hong Kong, Lindsay C. Stringer, Jouni Paavola.

**Data curation:** Zheng-Hong Kong, Lindsay C. Stringer, Jouni Paavola.

**Formal analysis:** Zheng-Hong Kong.

**Investigation:** Zheng-Hong Kong.

**Methodology:** Zheng-Hong Kong.

**Project administration:** Zheng-Hong Kong.

**Resources:** Zheng-Hong Kong, Lindsay C. Stringer.

**Software:** Zheng-Hong Kong.

**Supervision:** Lindsay C. Stringer, Jouni Paavola.

**Validation:** Zheng-Hong Kong.

**Visualization:** Lindsay C. Stringer.

**Writing – original draft:** Zheng-Hong Kong.

**Writing – review & editing:** Zheng-Hong Kong, Lindsay C. Stringer, Jouni Paavola.

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
