## [Decision Letter · Decision Letter 0]

19 Dec 2022

PONE-D-22-30804Knowledge exchange in the implementation of National Environmental Programmes (NEPs) in China: A complex picturePLOS ONE

Dear Dr. Kong,

Thank you for submitting your manuscript to PLOS ONE. After careful consideration, we feel that it has merit but does not fully meet PLOS ONE’s publication criteria as it currently stands. Therefore, we invite you to submit a revised version of the manuscript that addresses the points raised during the review process.

We look forward to receiving your revised manuscript.

Kind regards,

Peter Edwards

Academic Editor

PLOS ONE

Journal Requirements:

**Additional Editor Comments:**

I would like to thank the authors for their manuscript.

The reviewers have noted that it is generally well written, but have highlighted a number of areas where improvements could be made. In particular, the methodology should be improved, and a thorough copy edit and language check should be conducted.

I would encourage the authors to carefully consider and respond to all of the reviewers comments.

Reviewers' comments:

Reviewer's Responses to Questions

**Comments to the Author**

1. Is the manuscript technically sound, and do the data support the conclusions?

Reviewer #1: Yes

Reviewer #2: Yes

2. Has the statistical analysis been performed appropriately and rigorously? 

Reviewer #1: N/A

Reviewer #2: No

3. Have the authors made all data underlying the findings in their manuscript fully available?

Reviewer #1: No

Reviewer #2: Yes

4. Is the manuscript presented in an intelligible fashion and written in standard English?

Reviewer #1: Yes

Reviewer #2: No

5. Review Comments to the Author

Reviewer #1: In general, the paper is well written and definitely sheds light on an interesting and under-researched issue (in the geographical area, and also the study participants). It was very interesting to read the important role of village heads and some issues around junior and frontline scientists.

Some comments are:

1) the research methodology is interesting and the paper goes into some detail about the thematic analysis of the interviews and survey details, and whilst the paper states that the results are presented in an actor-oriented strategy, I really think that some more description of the themes and survey results, perhaps in a couple of tables and/or charts, would have benefitted the paper a lot.

2) The charts in the paper were very basic and could have had more detail and it would have been useful to have reported some simple non-random statistical analysis, such as crosstabes

3) The heading hierarchy in the article is confusing (i.e. page 7), and there are far too many sub-levels of headings, which are not reflected in the text

4) There is also inconsistent use of text formatting, such as bold, italics, quotation marks etc.

5) Some in-text references (i.e. page 4 but also other pages) give hyperlinks instead of correctly formatted references

6) The order given in section 3.1.1.3 regarding the ages of the farmers is given in a strange order (either report these results by ages or percentage in ascending or descending order)

7) The numbers next to the rather extensive list of quotes are rather confusing, it would have been useful to give a simple code to help with reading, such as "Farmer, 46" or similar

In conclusion, some more detail on the themes and survey responses/analysis other than just part of the text would really benefit the paper; as well as fixing the minor issues regarding formatting, heading hierarchy etc.

Reviewer #2: For the Last question above, I encourage the authors to run some copyediting of the manuscript.

See further my full review enclosed in the attached document. This includes my general comments and some more detailed comments on each of the Sections of the manuscript.

6. PLOS authors have the option to publish the peer review history of their article (what does this mean?). If published, this will include your full peer review and any attached files.

Reviewer #1: No

Reviewer #2: No

---

## [Author Response · Author response to Decision Letter 0]

15 Feb 2023

We greatly appreciate the comments and suggestions from the 2 reviewers. To address them, we have prepared a complete response letter. Please find details in the rebuttal letter.

---

## [Decision Letter · Decision Letter 1]

21 Jun 2023

PONE-D-22-30804R1Knowledge exchange in the implementation of National Environmental Programmes (NEPs) in China: A complex picturePLOS ONE

Dear Dr. Kong,

Thank you for submitting your manuscript to PLOS ONE. After careful consideration, we feel that it has merit but does not fully meet PLOS ONE’s publication criteria as it currently stands. Therefore, we invite you to submit a revised version of the manuscript that addresses the points raised during the review process.

The reviewers are overall happy with the revisions that you have made. They have suggested a number of very minor edits - typographical errors and some points of clarification or definition. Please review the suggested edits and make any required minor amendments.

We look forward to receiving your revised manuscript.

Kind regards,

Peter Edwards

Academic Editor

PLOS ONE

Journal Requirements:

Reviewers' comments:

Reviewer's Responses to Questions

**Comments to the Author**

1. If the authors have adequately addressed your comments raised in a previous round of review and you feel that this manuscript is now acceptable for publication, you may indicate that here to bypass the “Comments to the Author” section, enter your conflict of interest statement in the “Confidential to Editor” section, and submit your "Accept" recommendation.

Reviewer #1: (No Response)

Reviewer #2: All comments have been addressed

2. Is the manuscript technically sound, and do the data support the conclusions?

Reviewer #1: Yes

Reviewer #2: Yes

3. Has the statistical analysis been performed appropriately and rigorously? 

Reviewer #1: Yes

Reviewer #2: Yes

4. Have the authors made all data underlying the findings in their manuscript fully available?

Reviewer #1: No

Reviewer #2: (No Response)

5. Is the manuscript presented in an intelligible fashion and written in standard English?

Reviewer #1: Yes

Reviewer #2: Yes

6. Review Comments to the Author

Reviewer #1: Thank you for the improvements, the manuscript is much improved. Some minor comments:

• P3, l65, specifies climate science, whereas the MS is not overtly about climate science

• P4, l75, street-level bureaucrats is not defined (is only done so later on) – should be clarified for an international audience

• P5, l103, insert ‘The’ before ‘Three’

• P5, l106, north-western is spelled with different initial capitals in another place in the document

• P5, from lines 111, might be worth a brief description on the levels of government in China

• P6, l 123, ‘afflicted’ is an incorrect term

• Methodology, ensure that is written consistently in the past tense (or the same tense throughout)

• Table 1 is poorly formatted

• P9, l201, clarify what is meant by ‘minors’

• Throughout the document, there is inconsistent use of quotes (either single, or double). Some words are also put in quotes unnecessarily (i.e. p33, l715, supply and demand)

• P13, l285, should be members

• Tables 3, 4 etc, the column with the text ‘person’ is confusing, not sure what it means

• P17, l379 etc, should Mandarin be capitalised?

• P18, l393, incorrect in-text reference

• Table 5, improve the formatting (justify text under the bullet points)

• P26, l563, ideally, do not start a sentence with ‘Actually’

A map showing the broad areas (the research stations) would be useful

Reviewer #2: (No Response)

7. PLOS authors have the option to publish the peer review history of their article (what does this mean?). If published, this will include your full peer review and any attached files.

Reviewer #1: No

Reviewer #2: No

---

## [Author Response · Author response to Decision Letter 1]

27 Jun 2023

Dear Reviewer,

Thank you very much for the constructive comments and suggestions that helped us to further improve the manuscript. Our responses to your feedback are outlined in more detail in the rebuttal letter.

We look forward to your response.

Kind regards,

Zheng-Hong, Lindsay, and Jouni

---

## [Editor Report · Decision Letter 2]

3 Jul 2023

Knowledge exchange in the implementation of National Environmental Programmes (NEPs) in China: A complex picture

PONE-D-22-30804R2

Dear Dr. Kong,

We’re pleased to inform you that your manuscript has been judged scientifically suitable for publication and will be formally accepted for publication once it meets all outstanding technical requirements.

Kind regards,

Peter Edwards

Academic Editor

PLOS ONE
---

## [Editor Report · Acceptance letter]

5 Jul 2023

PONE-D-22-30804R2 

Knowledge exchange in the implementation of National Environmental Programmes (NEPs) in China: a complex picture 

Dear Dr. Kong:

I'm pleased to inform you that your manuscript has been deemed suitable for publication in PLOS ONE. Congratulations! Your manuscript is now with our production department. 

Kind regards, 

on behalf of

Dr. Peter Edwards 

Academic Editor

PLOS ONE